# Lifestyle Changes Reduced Estimated White Matter Hyperintensities Based on Retinal Image Analysis

**DOI:** 10.3390/ijerph20043530

**Published:** 2023-02-16

**Authors:** Maria Lai, Jack Lee, Xinxin Li, Chloe Kwok, Marc Chong, Benny Zee

**Affiliations:** 1Centre for Clinical Research and Biostatistics, Jockey Club School of Public Health and Primary Care, The Chinese University of Hong Kong, Hong Kong SAR, China; 2Centre for Clinical Trials and Biostatistics Lab, CUHK Shenzhen Research Institute, Shenzhen 518057, China

**Keywords:** lifestyle questionnaire, automatic retinal image analysis, small vessel disease, dementia

## Abstract

This study evaluates if there is an association between lifestyle changes and the risk of small vessel disease (SVD) as measured by cerebral white matter hyperintensities (WMH) estimated by the automatic retinal image analysis (ARIA) method. We recruited 274 individuals into a community cohort study. Subjects were assessed at baseline and annually with the Health-Promoting Lifestyle Profile II Questionnaire (HPLP-II) and underwent a simple physical assessment. Retinal images were taken using a non-mydriatic digital fundus camera to evaluate the level of WMH estimated by ARIA (ARIA-WMH) to measure the risk of small vessel disease. We calculated the changes from baseline to one year for the six domains of HPLP-II and analysed the relationship with the ARIA-WMH change. A total of 193 (70%) participants completed both the HPLP-II and ARIA-WMH assessments. The mean age was 59.1 ± 9.4 years, and 76.2% (147) were women. HPLP-II was moderate (Baseline, 138.96 ± 20.93; One-year, 141.97 ± 21.85). We observed a significant difference in ARIA-WMH change between diabetes and non-diabetes subjects (0.03 vs. −0.008, respectively, *p* = 0.03). A multivariate analysis model showed a significant interaction between the health responsibility (HR) domain and diabetes (*p* = 0.005). For non-diabetes subgroups, those with improvement in the HR domain had significantly decreased in ARIA-WMH than those without HR improvement (−0.04 vs. 0.02, respectively, *p* = 0.003). The physical activity domain was negatively related to the change in ARIA-WMH (*p* = 0.02). In conclusion, this study confirms that there is a significant association between lifestyle changes and ARIA-WMH. Furthermore, increasing health responsibility for non-diabetes subjects reduces the risk of having severe white matter hyperintensities.

## 1. Introduction

Hong Kong faces an increasing demand for health care and social services for the rapidly ageing population. According to the Hong Kong Population Projections 2012–2041, the proportion of persons older than 65 years was only 3.3% in 1966, 12.4% in 2006 and rapidly increased to 14% in 2011. It is projected to increase to 31% in 2036. According to the Population ageing trend of Hong Kong, the median age of the Hong Kong population will increase from 45.3 years old in 2018 to 51.5 years old in 2038. In terms of the absolute number of people aged 65 or above, there were 1.27 million in 2018. One of the significant challenges for the ageing population is the continuously increasing size of the population with cognitive decline and dementia [1].

Among individuals in the community aged 65 or above, around 30% have severe cerebral white matter hyperintensities (WMH) as measured by magnetic resonance imaging (MRI) [2]. Severe WMH individuals will likely develop cognitive dysfunction and dementia in a couple of years [3]. Therefore, the number of people with cognitive health problems will increase dramatically. Severe WMH found in healthy elderly individuals can be treated with early preventive intervention. Recent studies have shown that the presence of WMH increases the future risks of dementia, stroke and death, and it may be used as a modifiable risk factor to mediate the risk of cognitive decline [4]. Recently, we have developed a machine-learning-based automatic retinal image analysis (ARIA) method to estimate the WMH based on retinal images alone, using WMH measured by MRI as the gold standard [2]. We further validated the method using another data set with MRI and confirmed the accuracy. We also suggested that retinal information may provide localisation of WMH [5]. This technology allows a cost-effective and accurate assessment of the severe WMH status for risk assessment. At the same time, there is evidence showing that lifestyle is a modifiable risk factor for disease prevention. One of the most apparent benefits of lifestyle changes is the change from a sedentary lifestyle to moderate physical activity [6]. Other lifestyle factors include diet, exercise, cognitive training, and vascular risk monitoring, with randomised trial evidence showing significant benefits in executive functioning, processing speed, and overall neuropsychological test battery score [7].

This study evaluates if there is an association between lifestyle changes and changes in the risk of estimated WMH using the machine-learning-based automatic retinal image analysis one year apart.

## 2. Methods

### 2.1. Study Design and Participants

A total of 274 individuals were recruited consecutively from the Centre for Clinical Research and Biostatistics of the Chinese University of Hong Kong. The inclusion criteria were as follows: Age ≥ 18 years, willingness to sign the informed consent, and willingness to comply with study procedures required in the protocol. The exclusion criteria included age < 18 years; poor retinal image quality; other eye diseases such as severe cataracts, glaucoma, atretopsia, and corneal plague, and subjects known to be distressed by a flashlight or have a photosensitive seizure. Only subject numbers and initials were indicated on the image record and questionnaire for data collection and analysis. The study has obtained approval from the Joint CUHK-NTEC Clinical Research Ethics Committee (CRE-2015.524) and complies with the Declaration of Helsinki principles.

### 2.2. Study Procedures

The design of the study is a community cohort study. Each potential subject was informed of the study objectives and overall requirements, followed by a detailed explanation of the consent form. The subject and investigators signed written informed consent if the subject agreed to participate in the study. Subjects were assessed at baseline and annually with the Health-Promoting Lifestyle Profile II Questionnaire (HPLP-II) and underwent a simple physical assessment, such as blood pressure, body weight, and height. In addition, their medical histories, such as hypertension, cholesterol, and diabetes status, were self-reported by participants. Retinal images were taken using a non-mydriatic digital fundus camera to evaluate the level of WMH estimated by ARIA.

### 2.3. Retinal Images

Retinal images were taken on both eyes using a non-mydriatic digital fundus camera (Canon CR2-AF). Retinal photographs were taken at 45 degrees on the optic disc, macular, temporal to but including the fovea, upper and lower temporal arcades, and nasal to the optic disc. The size of the retinal image was at least 768 × 576 pixels.

### 2.4. WMH Evaluation Using ARIA

ARIA-WMH is a screening tool for the level of WMH that is noninvasive, convenient, fast, and less labour-intensive, which can be widely used in a community setting. The statistical methods for automatic retinal imaging analysis can be found in previous publications [8,9,10,11]. ARIA uses a machine learning technique to determine the risk assessment model. Previous validation studies have been done to confirm the accuracy of estimating the volume and grading of WMH [2]. The methods include fractal analysis, high-order spectra analysis, and statistical texture analysis, which target specific characteristics of the retinal image. The risk of WMH is a score from 0 to 1, representing the probability of having severe grading of WMH as if it was measured using magnetic resonance imaging (MRI) as a gold standard. In the community application, an ARIA-WMH score of less than 0.4 is considered normal. A score of 0.4–0.6 represents a moderate risk level, and more than 0.6 is defined as a high-risk level.

### 2.5. Lifestyle Changes Assessment Using HPLP-II

Lifestyle was assessed by the Health-Promoting Lifestyle Profile II Questionnaire (HPLP-II). This instrument has been widely used in different settings. It was initially developed by Walker, Sechrist and Pender (1987) and later revised as the HPLP-II [12]. In 1997, it was translated into a Chinese (Taiwanese) version and verified the translated version’s reliability and validity [13,14,15]. HPLP-II contains 52 items and 6 domains consisting of 8 or 9 items. The domains include interpersonal relations (IR), spiritual growth (SG), nutrition (NU), stress management (SM), health responsibility (HR), and physical activity (PA). Items are measured using a four-point Likert scale (1 = never, 2 = sometimes, 3 = often, and 4 = routinely), with higher scores indicating healthier lifestyles. The total score ranges from 52 to 208, divided into 4 grades (52–90, 91–129, 130–168, and 169–208, indicating poor, moderate, good, and excellent levels, respectively). The mean item score ranges from 1 to 4 [16,17].

### 2.6. Statistical Analysis

Data were expressed as mean ± SD or N (%) in frequency tables. To analyse the relationship between participant characteristics and the ARIA-WMH change, we first calculated each individual’s change scores of ARIA-WMH risks. We then compared them between groups within each participant’s characteristics using a 2-sample *t*-test. For continuous variables such as the HPLP-II domains, we used a 1-sample *t*-test to assess the significance of the changes. In addition, multivariate regression analysis with a stepwise method and subgroup analysis was utilised to analyse the associations between lifestyle changes and ARIA-WMH changes. The SAS system was used for statistical analysis. *p* < 0.05 was considered statistically significant. The sample size requirement was based on a correlation test. We postulated that we would obtain a correlation coefficient of at least 0.3 or higher between changes in WMH and in HPLP-II domains. Therefore, we will have 95% power using a 5% 2-sided test if we enter at least 140 subjects in this part of the study. Since the overall sample size of this study was 193, and the non-diabetic subgroup was 160 subjects. Therefore, we have more than 95% power to detect the postulated associations.

## 3. Results

### 3.1. Participant Characteristics

Of the 274 eligible participants, 193 (70%) completed the HPLP-II and ARIA-WMH assessments at baseline and after one year of follow-up. Demographic data are shown in Table 1. The mean age was 59.1 ± 9.4 years. Among the subjects, 76.2% (147) were women, 31.4% (60) were overweight and obese, 42.9% (82) had a high waist-hip ratio, 38.3% (74) had hypertension, 14.4% (27) had diabetes mellitus, 58% (109) had a family history of cardiovascular disease (CVD), 28.2% (51) treated for abnormal cholesterol, 2.6% (5) were smokers, and 21.8% (42) were drinkers.

### 3.2. ARIA-WMH Scores

Table 2 shows baseline and one-year ARIA-WMH scores and the changes in the ARIA-WMH during the same period. At baseline, 61% (118) of individuals were at a low-risk level, 33% (64) at a moderate-risk level, and 6% (11) at a high-risk level. The distribution was similar to the distribution at 1 year. Furthermore, the ARIA-WMH changes were quite symmetrical, with <−0.1 (n = 29, 15.1%), −0.1–0 (n = 67, 34.7%), 0–0.1 (n = 67, 34.7%), and >0.1 (n = 30, 15.5%) respectively. Overall, these results indicate that based on the ARIA-WMH of participants in the community, about 30–33% of subjects have a moderate risk, and only 5–8% have a high risk.

### 3.3. HPLP-II Scores

The baseline and one-year HPLP-II total scale and six subscales are shown in Table 3. At baseline, the total scores were 138.96 ± 20.93, and the mean item score was 2.67 ± 0.40. According to the HPLP-II subscales, interpersonal relations had the highest score (2.81 ± 0.45), followed by spiritual growth, nutrition, stress management, and health responsibility, while physical activity had the lowest score (2.49 ± 0.62). At 1 year, the average total HPLP-II score significantly increased as compared to the baseline, with a mean change score of 0.06 (*p* < 0.05). Specific domains that had significant changes as compared with the baseline scores include interpersonal relations (mean change score for IR = 0.05, *p* < 0.05), spiritual growth (mean change score of SG = 0.06, *p* < 0.05), nutrition (mean change score of NU = 0.07, *p* < 0.05) and health responsibility (mean change score of HR = 0.08, *p* < 0.05). The mean scores for stress management and physical activity domains did not change significantly within 1 year. Overall, these results suggested that the participants in this cohort significantly improved overall HPLP-II and various HPLP-II domains.

### 3.4. Univariate Analysis of ARIA-WMH

The change in the ARIA-WMH score between baseline and one year was compared according to participant characteristics (Table 1). If the ARIA-WMH change score was positive, it means that the WMH increased; otherwise, if the ARIA-WMH change score was negative, it meant that the WMH decreased. A significant difference in ARIA-WMH change between diabetes and non-diabetes subjects (0.03 ± 0.08 vs. −0.008 ± 0.13, respectively, *p* = 0.03). No significant difference was found according to age, gender, BMI, Waist-Hip Ratio, hypertension, family history of CVD, treatment for cholesterol, smoking, and drinking.

### 3.5. Association between Changes in ARIA-WMH and HPLP-II Domains

The changes in the six domains of HPLP-II between baseline and one year were calculated and divided into two categories. An increased score was treated as a positive change group, meaning participants adopted a healthier lifestyle. In contrast, a decrease or no change in the score was treated as a negative change group, meaning they adopted a poorer lifestyle. In a univariate regression model for predicting ARIA-WMH change, only HR and PA domains were significant predictors among HPLP-II domains and personal characteristics, including age, gender, BMI, Waist-Hip Ratio, hypertension, diabetes mellitus, family history of CVD, treatment for cholesterol, smoking, and drinking.

We used the stepwise multivariate regression method to examine which HPLP-II domains were associated with ARIA-WMH change. We entered all six dimensions into the regression model as independent variables and assessed potential interactions between the domains and the corresponding personal characteristics. The final stepwise regression model showed a significant interaction between the HR domain and diabetes (*p* = 0.01). In addition, the PA domain was also negatively related to the change in ARIA-WMH (*p* = 0.02).

We further examined the interaction effect between the HR domain and diabetes to understand how HR change affects ARIA-WMH in the diabetes and non-diabetes subgroups (Table 4). For non-diabetes subgroups, those with a positive change in the HR domain had a decrease in ARIA-WMH, and those with a negative change in HR had a slight increase in ARIA-WMH. The difference between the groups with a positive change versus a negative change of HR was statistically significant (−0.04 ± 0.12 vs. 0.02 ± 0.13, respectively, *p* = 0.003). On the other hand, no significant difference was found in the comparison of positive and negative changes in HR in the diabetes subgroup.

## 4. Discussion

Our study indicated a significant association between diabetes and ARIA-WMH, consistent with previous studies [18,19,20,21,22]. Based on MRI data, Espeland et al. (2016) found that type 2 diabetes was associated with increased WMH [20]. They found that women with diabetes had slightly greater loss of total brain volumes and significant increases in total ischemic lesion volumes relative to those without diabetes. Diabetes was associated with lower scores in global cognitive function and its subdomains. Schneider et al. (2017) showed that subjects with diabetes with HbA1c ≥ 7.0% had smaller total and regional brain volumes and an increased burden of WMH (all *p* < 0.05) compared with those with HbA1c < 7.0% [22]. They also found those with a longer duration of diabetes (≥10 years) had smaller brain volumes and a higher burden of lacunes (all *p* < 0.05) than those with a diabetes duration of <10 years. These results support our observation that diabetes is significantly associated with increased ARIA-WMH (*p* = 0.03). The impact of diabetes is so strong that none of the lifestyle domains could have a positive impact that can reverse the potential damage shown in ARIA-WMH. Hence, it is reasonable to see a significant interaction between diabetes and health responsibility showing that health responsibility was significantly associated with a reduction in ARIA-WMH among the non-diabetes subjects. Furthermore, there was randomized clinical trial evidence showing that lifestyle changes affect not only white matter hyperintensities but also cognitive performance. For example, the FINGER trial showed that multidomain lifestyle intervention could improve cognitive performance. It is reasonable to expect an effective change in the health responsibility domain to achieve a reduction in the risk of ARIA-WMH and eventually translate into cognitive health benefits [7].

In 2019, the World Health Organization (WHO) published “WHO Guidelines on risk reduction of cognitive decline and dementia.” It provides evidence-based recommendations on lifestyle and behavioural interventions to delay or prevent cognitive decline and dementia. Potential preventive interventions include physical activity, nutrition intervention, social activity, weight management, and intellectual activity. Our study also found that improved physical activity is beneficial for reducing ARIA-WMH. Evidence shows that physical activity has a lower risk of cognitive impairment from a longitudinal population-based study with a follow-up duration of 9 years. The results show that people who regularly exercised had a lower risk of developing dementia than those who did not exercise regularly [23].

Some limitations of the study should be noticed when interpreting the results. Firstly, the subjects who participated in this study were volunteers who may be more health-conscious than the average person in the community. However, after the first assessment, they were exposed to the HPLP-II questionnaire and increased their health status awareness. Hence the average HPLP-II total change score significantly increased one year after the baseline. Other limitations include the small sample size for subgroup analysis, which limited the statistical power of the model when we analysed the risk factors of ARIA-WMH. In the future, we may develop lifestyle interventions along the line of multidomain activities, which is, to a certain extent, reflected in the health responsibility domain.

## 5. Conclusions

In this cohort study, we found that diabetes is significantly associated with an increased risk of severe white matter hyperintensities (WMH) as measured by ARIA-WMH. 2. We have demonstrated that lifestyle changes may modify WMH. For people without diabetes, improving health responsibility could reduce the risk of severe WMH. However, no improvement in risk of severe WMH was found in the diabetes subgroup. In general, promoting physical activity significantly reduces the risk of severe WMH.

## Figures and Tables

**Table 1 ijerph-20-03530-t001:** Participant characteristics and ARIA-WMH change.

	N	%	ARIA-WMH Change		
			Mean	SD	*p*
Gender					
Female	147	76.2	−0.002	0.12	0.94
Male	46	23.8	−0.004	0.13	
Age					
<65	139	72.0	−0.004	0.13	0.82
≥65	54	28.0	0.0004	0.10	
BMI (kg/m^2^)					
Desirable	131	68.6	−0.012	0.13	0.08
Overweight and Obese	60	31.4	0.018	0.10	
Missing	2				
Waist-hip ratio					
Normal	109	57.1	−0.008	0.14	0.42
High	82	42.9	0.006	0.10	
Missing	2				
Hypertension					
No	119	61.7	−0.005	0.12	0.77
Yes	74	38.3	0.0005	0.12	
Diabetes Mellitus					
No	160	85.6	−0.008	0.13	0.03 *
Yes	27	14.4	0.034	0.08	
Missing	6				
Family history of CVD					
No	79	42.0	−0.003	0.14	0.94
Yes	109	58.0	−0.002	0.11	
Missing	5				
Tx for cholesterol					
No	130	71.8	−0.007	0.13	0.35
Yes	51	28.2	0.010	0.10	
Missing	12				
Smoking					
Non-Smoker	188	97.4	−0.003	0.12	0.12
Smoker	5	2.6	0.025	0.03	
Drinking					
Non-drinker	151	78.2	−0.009	0.12	0.17
Drinker	42	21.8	0.021	0.13	

BMI, body mass index; BMI Desirable, <25 kg/m^2^; BMI Overweight and Obese, ≥25 kg/m^2^; Waist-hip ratio (WHR) Normal, WHR ≤0.85 (Female) or WHR ≤ 1 (Male); Waist-hip ratio High, WHR >0.85 (Female) or WHR >1 (Male); Hypertension, SBP ≥130 mmHg or DBP ≥80 mmHg. * *p* < 0.05.

**Table 2 ijerph-20-03530-t002:** Baseline and One-year ARIA-WMH Scores (N = 193).

ARIA-WMH Scores	Baseline	One-Year
	N (%)	N (%)
Mean ± SD	0.344 ± 0.17	0.341 ± 0.17
<0.4	118 (61.1)	118 (61.1)
0.4~0.6	64 (33.2)	59 (30.6)
>0.6	11 (5.7)	16 (8.3)
ARIA-WMH Change Score	N	%
Mean ± SD	−0.003	0.12
<−0.1	29	15.1
−0.1~0.01	67	34.7
0~0.1	67	34.7
>0.1	30	15.5

**Table 3 ijerph-20-03530-t003:** Baseline and One-year HPLP-II Scores (N = 193, Mean ± SD).

Scale and Subscales	Question Number	Baseline		One-Year		Mean Item Score Change
		Total Scores	Mean Item Score	Total Scores	Mean Item Score	
Total	52	138.96 ± 20.93	2.67 ± 0.40	141.97 ± 21.85	2.73 ± 0.42	0.06 ± 0.019 *
IR	9	25.32 ± 4.08	2.81 ± 0.45	25.77 ± 4.22	2.86 ± 0.47	0.05 ± 0.024 *
SG	9	25.07 ± 4.79	2.79 ± 0.53	25.63 ± 4.99	2.85 ± 0.55	0.06 ± 0.026 *
NU	9	24.82 ± 3.98	2.76 ± 0.44	25.42 ± 3.76	2.82 ± 0.42	0.07 ± 0.022 *
SM	8	21.19 ± 3.79	2.65 ± 0.47	21.53 ± 3.90	2.69 ± 0.49	0.04 ± 0.027
HR	9	22.67 ± 4.22	2.52 ± 0.47	23.36 ± 4.50	2.60 ± 0.50	0.08 ± 0.029 *
PA	8	19.89 ± 4.95	2.49 ± 0.62	20.27 ± 4.90	2.53 ± 0.61	0.05 ± 0.028

IR, interpersonal relations; SG, spiritual growth; NU, nutrition; SM, stress management; HR, health responsibility; PA, physical activity. * *p* < 0.05.

**Table 4 ijerph-20-03530-t004:** Association of changes in ARIA-WMH and HR domain by Diabetes subgroup.

Diabetes Subgroup	HR	N	%	ARIA-WMH Change		
				Mean	SD	*p*
Non-diabetes	negative	85	53.13	0.02	0.13	0.003
	positive	75	46.88	−0.04	0.12	
Diabetes	negative	12	44.44	0.04	0.09	0.691
	positive	15	55.56	0.05	0.05	

## Data Availability

The data presented in this study are comprised in the manuscript.

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
