# Peer review of "Lifestyle Changes Reduced Estimated White Matter Hyperintensities Based on Retinal Image Analysis"

_ijerph, 2023, doi:10.3390/ijerph20043530_

Round 1
Reviewer 1 Report
The authors approach a very important and pertinent topic related with the use of a putative biomarker (retinal image analysis) that predicts the presence of WMH in a community-based population and its association with life-style characteristics and changes. A sample of community-dwelling individuals were evaluated at baseline and one year after, seeking to determine if changes in life-style are reflected in changes of estimated WMH employing retinal image analysis. Nevertheless, there are significant issues that need to be improved for publication.
Abstract: the conclusion should be rephrased according to the study´s results.
Introduction:
First paragraph: reference #1 should be placed at the end of the paragraph if all the information came from this reference. If not, the reference associated with the second part of this paragraph must be included.
Methods.
Patient recruitment criteria are not quite clear to me. The authors only state that it was a community cohort study, but there is no information as to where the universe of the patients came from and how they were selected (consecutively included, random inclusion or other)? How many subjects did you screen for the study at the beginning and how many did you exclude and why? Please clarify the recruiting procedure.
Results.
Table 2: It is not clear in which group those patients whose change was 0 were classified (-0.1 – 0 or 0 – 0.1).
Table 3. the mean change for HPLP-II item scores at baseline and one year have standard deviations which are much larger than the means, is that correct, because it would imply an enormous variability? Can that be significant? Please check if this is correct. I am not sure which statistical test the authors employed to compare changes between baseline and one year evaluations. I suggest the authors compare baseline and one year evaluations using a t-test for dependent variables (because each subject has two readings for each variable). Maybe that is what they did, but it is not clear for me, because in the Methods section they state the used one-sample and 2-sample t-tests. Please clarify this.
Table 4. I do not understand the explanation in the manuscript for the results presented in Table 4. Why does the non-diabetes group only have negative HR and the Diabetes group has two positive and one negative HR? Please clarify the table and the manuscript text.
Discussion:
The Discussion starts by giving the HPLP scores and item scores for the general population in Hong Kong. The authors cannot consider their results as representative of the population of Hong Kong; they would need to carry out a much more complex sampling procedure, which they do not explain at all.
The authors express in the Introduction that the aim of the study was to evaluate lifestyle changes and changes in the risk of estimated WMH employing automated retinal image analysis. Nevertheless, in the Discussion this is not clearly analyzed. They mainly refer to the detailed characteristics of the cohort´s lifestyle at baseline evaluated through the HPLP score and the association of diabetes with increased ARIA-WMH score. The association of lifestyle changes with estimated WMH changes is not discussed at all. Too much space is given to describe other authors´ results.
In the abstract the authors conclude that ARIA-WMH can be used to estimate SVD. This was not expressed in the Discussion, but it is totally incorrect. This study does not evaluate if ARIA-WMH can estimate SVD, this was apparently demonstrated by the authors in 2 previous studies (references 2 and 5). The experimental design does not allow evaluating this because they lacked brain MRI images to compare WMH burden with ARIA-WMH measurements.
In general, I think the Discussion is very poor considering the great amount of results presented in the manuscript, and should be totally rewritten.
Reviewer 2 Report
The article presents the results of research on the relationship between lifestyle changes and the risk of developing small vessel disease. The authors followed 274 patients. They checked their retinal image and assessed it using the Health Promoting Lifestyle Profile II (HPLP-II) questionnaire and a simple clinical assessment. Below, in points, I present my questions, doubts and comments. Please respond to all the issues raised. 1. Line 101 states that patients self-reported information about their test results, such as blood pressure. There is a high risk that due to an error in the readings (if the tests were performed in-house) the data entered into the test will be unreliable. How exactly were the results referred to in this row retrieved in this case? 2. Uneven distribution of cockroaches is a cause for concern. Such research cannot be applied to the whole of society. Too large a disparity between the cockroaches included in the study. What caused the disparity? How does the number of patients compare to the local community? Especially since the discussion is about Hong Kong's population. 3. Especially since Hong Kong's population is being discussed. 4. The whole article is very imitative and adds nothing new. What have you discovered during your research that you didn't know before about the topic discussed in the article?Author Response
Please see the attachment.

Round 2
Reviewer 2 Report
Thank you for correcting the text.
Nevertheless, I still maintain that the article adds nothing to the state of knowledge and is reproducible and repetitive. There are many articles on a similar subject.